# Coke-Resistant Ni/CeZrO_2_ Catalysts for Dry Reforming of Methane to Produce Hydrogen-Rich Syngas

**DOI:** 10.3390/nano12091556

**Published:** 2022-05-04

**Authors:** Intan Clarissa Sophiana, Ferry Iskandar, Hary Devianto, Norikazu Nishiyama, Yogi Wibisono Budhi

**Affiliations:** 1Department of Chemical Engineering, Faculty of Industrial Technology, Institut Teknologi Bandung, Bandung 40132, Indonesia; intanclarissas@gmail.com (I.C.S.); hardev@che.itb.ac.id (H.D.); 2Research Center for Nanoscience and Nanotechnology, Institut Teknologi Bandung, Bandung 40132, Indonesia; 3Department of Physics, Faculty of Mathematics and Natural Science, Institut Teknologi Bandung, Bandung 40132, Indonesia; ferry@fi.itb.ac.id; 4Department of Chemical Engineering, Engineering Science School, Osaka University, Osaka 565-0871, Japan; nisiyama@cheng.es.osaka-u.ac.jp

**Keywords:** nickel, ceria-zirconia, coke-resistant catalysts, dry reforming of methane, greenhouse gases, syngas production

## Abstract

Dry reforming of methane was studied over high-ratio zirconia in ceria-zirconia-mixed oxide-supported Ni catalysts. The catalyst was synthesized using co-precipitation and impregnation methods. The effects of the catalyst support and Ni composition on the physicochemical characteristics and performance of the catalysts were investigated. Characterization of the physicochemical properties was conducted using X-ray diffraction (XRD), N_2_-physisorption, H_2_-TPR, and CO_2_-TPD. The results of the activity and stability evaluations of the synthesized catalysts over a period of 240 min at a temperature of 700 °C, atmospheric pressure, and WHSV of 60,000 mL g^−1^ h^−1^ showed that the 10%Ni/CeZrO_2_ catalyst exhibited the highest catalytic performance, with conversions of CH_4_ and CO_2_ up to 74% and 55%, respectively, being reached. The H_2_/CO ratio in the product was 1.4, which is higher than the stoichiometric ratio of 1, indicating a higher formation of H_2_. The spent catalysts showed minimal carbon deposition based on the thermo-gravimetry analysis, which was <0.01 gC/gcat, so carbon deposition could be neglected.

## 1. Introduction

In recent decades, the abundances of methane (CH_4_) and carbon dioxide (CO_2_) in natural gas reserves and the atmosphere have attracted the attention of researchers for use in the production of value-added products, which would reduce global warming by mitigating the emission of these greenhouse gases [1]. Natural gas utilization often involves reforming technology that converts natural gas into synthesis gas (also known as “syngas”). Reforming technologies include steam, partial oxidative, and dry reforming [2,3]. Steam reforming involves the reaction of CH_4_ and steamed water (H_2_O), whereas partial oxidative reforming involves the reaction of CH_4_ and oxygen (O_2_), where the moles of O_2_ required are less than the stoichiometric value [4,5]. Dry reforming of methane (DRM) involves the reaction of the greenhouse gases CH_4_ and CO_2_ to produce syngas containing hydrogen (H_2_) and carbon monoxide (CO) at a H_2_/CO ratio of approximately 1 [6,7,8,9]. The syngas produced in the DRM can be used as a starting material for the manufacture of dimethyl ether, acetic acid, and alcohols via the synthesis of oxo-alcohols as an intermediate or final product in the petrochemical industry [10,11].

The DRM is reversible and endothermic; therefore, a large amount of energy is required to make the process feasible [12]. The DRM requires a catalyst that plays a role in maximizing the production of syngas by increasing the rate of chemical reactions [13]. During the DRM, a reverse water–gas shift reaction (WGSR) occurs, which produces water (H_2_O) to encourage the endothermic steam reforming of methane (SRM) and exothermic steam/carbon gasification as a side reaction. In addition, other side reactions can occur during the DRM: The CO produced via DRM, reverse WGSR, and steam/carbon gasification can encourage the Boudouard reaction, while the CH_4_ in the feed can decompose into carbon and H_2_. The Boudouard reaction and methane decomposition produce solid carbon, which can cover the active sites on the surface of the catalyst, resulting in catalyst deactivation. The chemical reactions that can occur during the DRM are shown in Equations (1)–(6) below [14].

DRM reaction:(1)$${\mathrm{CH}}_{4}+{\mathrm{CO}}_{2}\;⇄\;2\mathrm{CO}+2H_{2}\;∆H{{}^{\circ}}_{298\;K}=+247\;\mathrm{kJ}/\mathrm{mol}$$

Reverse WGSR:(2)$${\mathrm{CO}}_{2}+H_{2}\;⇄\;\mathrm{CO}+H_{2}O\;∆H{{}^{\circ}}_{298\;K}=+41\;\mathrm{kJ}/\mathrm{mol}$$

SRM reaction:(3)$${\mathrm{CH}}_{4}+H_{2}O\;⇄\;\;\mathrm{CO}+3H_{2}\;∆H{{}^{\circ}}_{298\;K}=+206\;\mathrm{kJ}/\mathrm{mol}$$

Boudouard reaction:(4)$$2\mathrm{CO}\;⇄\;C_{\left(s\right)}+{\mathrm{CO}}_{2}\;∆H{{}^{\circ}}_{298\;K}=-172\;\mathrm{kJ}/\mathrm{mol}\;$$

Methane decomposition reaction:(5)$${\mathrm{CH}}_{4}\;⇄\;C_{\left(s\right)}+2H_{2}\;∆H{{}^{\circ}}_{298\;K}=+74\;\mathrm{kJ}/\mathrm{mol}\;$$

Steam/carbon gasification reaction:(6)$$C_{\left(s\right)}+H_{2}O\;⇄\;\mathrm{CO}+H_{2}\;∆H{{}^{\circ}}_{298\;K}=+131\;\mathrm{kJ}/\mathrm{mol}$$

Precious metal-based catalysts, such as Pt, Rh, Ru, Pd, and Ir, exhibit high catalytic performance in the DRM and high resistance to carbon deposition on the catalyst surface [15,16,17,18,19,20,21,22,23,24,25]. However, owing to the lack of precious metal resources and high prices, many researchers are focusing on transition metals, such as Ni and Co, which have lower prices [26,27,28,29,30,31,32,33,34,35,36]. Based on the catalytic activity of transition-metal-based catalysts in the DRM, Ni is the only transition metal that is comparable with noble metals [4].

Although Ni-based catalysts have high catalytic activity, these can undergo catalyst deactivation during the DRM via two mechanisms: (1) Carbon deposition that occurs within the temperature range of 500–700 °C, resulting in the blockage of surface active sites, and (2) sintering that occurs within the temperature range of 700–800 °C, which results in a loss of active surface area [31]. The lack of stability of Ni-based catalysts in the DRM may limit their commercial use; therefore, Ni-based catalysts must be modified to improve their catalytic performance and resistance to carbon deposition [4].

The support and promoter materials used in DRM catalysts generally have a high oxygen storage capacity (OSC), which when combined with the active sites of the catalyst, can increase the mobility of oxygen atoms through the crystal lattice and promote the formation of oxygen site vacancies in the catalyst [37]. Several promoters, such as ceria (CeO_2_), have been added to conventional Ni/Al_2_O_3_ catalysts to increase their reduction/oxidation (redox) potential [38,39]. As a result, adding a promoter to conventional catalysts can increase catalytic performance in the DRM [40]. Therefore, the addition of CeO_2_, which has a high OSC, to the DRM catalyst on various supports, such as alumina, silica, and zirconia, can improve the catalytic performance.

The use of a catalyst support with weak Lewis acid sites and/or the presence of alkaline sites can improve the performance, stability and coke-resistance of the DRM catalyst, of which the ZrO_2_ support fits these criteria [41]. Several researchers have used CeZr as a catalyst support for the DRM [42,43,44,45,46,47,48,49]. Kambolis et al. [48] showed that the use of a ceria-zirconia catalyst support with a high zirconia content (28 wt.% ceria and 72 wt.% zirconia) resulted in high product yields. To date, research using ceria-zirconia-supported Ni catalysts with a high zirconia:ceria ratio and varying Ni contents has not been reported; therefore, it has the potential to be investigated to obtain the optimum catalyst for the DRM.

## 2. Materials and Methods

### 2.1. Materials

Cerium(III) nitrate hexahydrate (Ce(NO_3_)_3_·6H_2_O, >99.99%), zirconium(IV) chloride (ZrCl_4_, >98%), nickel(II) nitrate hexahydrate (Ni(NO_3_)_2_·6H_2_O, >99%), and ammonium hydroxide solution (NH_4_OH, 25 wt.%) were used in this study and purchased from Merck (Damstadt, Germany). Demineralized water was used throughout the synthesis process. A mixture of CH_4_ and CO_2_ (50%:50%) was used for the catalyst evaluation experiments. The gases H_2_ (99.95%), N_2_ (99.99%), and Ar (99.9%) were used as the reducing agent, diluent and inert internal standard for the reaction, and carrier gas for the gas chromatographic (GC) analysis, respectively. All the materials were used without further purification.

### 2.2. Preparation of Catalysts

#### 2.2.1. Catalyst Based on the ZrO_2_ Support

ZrO_2_ was prepared via precipitation. ZrCl_4_ was dissolved in water to form a 0.5 M solution, which was precipitated with a basic solution containing 25 wt.% NH_4_OH at pH 9 and a temperature of 60 °C, and then aged by stirring. The precipitate was washed several times using deionized water, dried at 105 °C, and calcined in air at 800 °C for 6 h. An aqueous solution of Ni(NO_3_)_2_·6H_2_O and/or Ce(NO_3_)_3_·6H_2_O was impregnated into the ZrO_2_ support via classical incipient wetness impregnation to form 5%NiO/ZrO_2_ and 5%NiO-10%CeO_2_/ZrO_2_. The resultant samples were dried at 105 °C overnight, and calcined in the air at 700 °C for 6 h. 

#### 2.2.2. Catalyst Based on the Ce_0.1_Zr_0.9_O_2_ Support

The Ce_0.1_Zr_0.9_O_2_ mixed metal oxide was prepared via co-precipitation. Ce(NO_3_)_3_·6H_2_O and ZrCl_4_ were dissolved in water to form a 0.5 M solution and co-precipitated with a basic solution containing 25 wt.% NH_4_OH at pH 9 and a temperature of 60 °C, and then aged by stirring. The precipitate was washed several times using deionized water, dried at 105 °C for overnight, and calcined in air at 800 °C for 6 h. The Ce_0.1_Zr_0.9_O_2_ (CeZrO_2_) support was impregnated via classical incipient wetness impregnation with an aqueous solution of Ni(NO_3_)_2_·6H_2_O, with different amounts of the Ni precursor. The resultant sample was dried at 105 °C overnight and calcined in air at 700 °C for 6 h to finally obtain 5%NiO/CeZrO_2_ and 10%NiO/CeZrO_2_.

### 2.3. Characterization of Catalysts

X-ray diffraction (XRD) was used to determine the crystallinity of the catalysts. XRD measurements on the synthesized catalyst were performed using a Philips X’Pert MDR diffractometer (Royal Dutch Philips Electronics Ltd., Amsterdam, The Netherlands) equipped with a Cu anode tube to produce Cu-Kα (λ = 1.5405 Å) X-rays and operated at 40 kV and 35 mA, using a scan rate of 20° min^−1^. The catalyst sample was placed in the cell of the diffractometer and irradiated with X-rays. The diffraction data were obtained within a 2θ range of 5–90°, with a scan speed of 4° min^−1^. 

The Brunauer–Emmett–Teller (BET) isotherm method was used to measure the mass-specific surface area of the catalyst sample based on N_2_ physisorption using the Micromeritics Tristar II instrument (Micromeritics Instrument Corp., Norcross, GA, USA). The solid powder to be analyzed was placed in the sample cell and weighed. The sample cells were degassed to remove water molecules and impurity gases adhering to the surface of the catalyst sample. This process was performed at 250 °C for 2 h. After the degassing, the catalyst sample was cooled to room temperature. The sample cells were transferred to the gas sorption analyzer section of the instrument and analyzed. The pore size distribution was determined using the Barrett–Joyner–Halenda (BJH) method.

Hydrogen temperature-programmed reduction (H_2_-TPR) was used to determine the interactions between metals. A Micromeritics Chemisorb 1750 instrument (Micromeritics Instrument Corp., Norcross, GA, USA) was used to carry out H_2_-TPR analysis. The catalyst (70 mg) was used to fill a U-shaped quartz tube, placed in a furnace, and degassed with Ar at 150 °C for 1 h. Thereafter, the sample was cooled to room temperature, and reduced under a flow rate of 40 mL min^−1^ of 5% H_2_ in Ar within the temperature range of 30–900 °C using a heating rate of 10 °C min^−1^. The TPR profile was recorded using a thermal conductivity detector (TCD), which monitored the consumption of H_2_ by the catalyst sample.

Carbon dioxide temperature-programmed desorption (CO_2_-TPD) was used to characterize the type of Lewis acid sites on the surface of the catalyst by measuring the number of CO_2_ molecules absorbed during heating at a constant temperature. During CO_2_-TPD, the sample is heated in an inert gas below a specific temperature, at which point the bond between the adsorbed gas and sample is broken, resulting in desorption of the gas. TPD measurements were carried out in a Micromeritics Chemisorb 2750 unit. The sample (70 mg) was introduced into a U-shaped quartz sample tube for each analysis. The probe gas used in the analysis was 5% CO_2_/He. Prior to the measurements, the catalyst sample was purged with He at 200 °C for 1 h to remove weakly bonded species. CO_2_ gas adsorption was carried out at 50 °C for 30 min using the 5% CO_2_/He gas mixture at a flow rate of 30 mL min^−1^ to ensure that the probe gas molecules saturated all sites on the surface of the catalyst. After adsorption, the sample was purged with He for 30 min. The desorption process was conducted between 50 and 900 °C, with a heating rate of 10 °C min^−1^. During the TPD stage, the concentration of the desorbed gas was measured using a TCD.

Thermogravimetric analysis (TGA) was used to measure the amount of carbon deposited on the surface (and in the pores) of the catalyst after the DRM. TGA, using the Shimadzu TA-60WS instrument (Shimadzu Corp., Kyoto, Japan), required a sample of approximately 5–10 mg to be placed in a cell and heated under an air-flow (50 mL min^−1^) from room temperature to 800 °C at a rate of 20 °C min^−1^.

### 2.4. Catalytic Performance Evaluation

Catalyst evaluation in the DRM was conducted at atmospheric pressure in a quartz fixed-bed reactor tube (ID = 10 mm) at 700 °C for 240 min. For each test, 60 mg (80–120 mesh) of the catalyst was packed between two ceramic wool plugs to keep the catalyst in a fixed position and diluted with 300 mg of SiC. First, the catalyst was purged with N_2_ at 400 °C for 1 h and reduced in a 1:1 H_2_:N_2_ gas mixture flowed at a rate of 100 mL min^−1^, while heating at a temperature of 700 °C for 2 h. Then, the system was purged with N_2_ at a flow rate of 100 mL min^−1^ for 30 min. The feed gas flow rate used for the activity and stability evaluations was regulated using a mass flow controller. The feed gas used was a mixture of CO_2_:CH_4_:N_2_ at a ratio of 1:1:1. The flow rate of the feed gas mixture was set at 60 mL min^−1^, resulting in a weight-hourly space velocity (WHSV) of 60,000 mL g^−1^ h^−1^. 

The products and unreacted gas feed were analyzed using online gas chromatography (Shimadzu GC-14B, Shimadzu Corp., Kyoto, Japan), which was carried out every 10 min, with the amount of sample analyzed being ca. 1 mL. After the feed through the reactor under the operating conditions was determined, the product stream from the reactor output was condensed to separate the H_2_O content caused by the reverse WGSR using a condenser. As a safety precaution, the H_2_O present in the reactor effluent must be condensed before the rest of the gas stream is analyzed in the online GC. The gas chromatograph was equipped with two columns, namely, Porapak-Q (PQ) and Molecular Sieve-5A (MS), arranged in parallel. The Porapak-Q column was used to analyze CO_2_, while the MS-5A column was used to analyze H_2_, N_2_, CH_4_, and CO. The conversion, yield, and H_2_/CO ratio were evaluated using Equations (7)–(11).
(7)$${\mathrm{CH}}_{4}\;\mathrm{conversion}\;\left(\%\right)=\frac{F_{{\mathrm{CH}}_{4,in}}-F_{{\mathrm{CH}}_{4,out}}}{F_{{\mathrm{CH}}_{4,in}}}\times 100\%$$
(8)$${\mathrm{CO}}_{2}\;\mathrm{conversion}\;\left(\%\right)=\frac{F_{{\mathrm{CO}}_{2,\;in}}-F_{{\mathrm{CO}}_{2,\;out}}}{F_{{\mathrm{CO}}_{2,in}}}\times 100\%$$
(9)$$H_{2}\;\mathrm{yield}\;\left(\%\right)=\frac{F_{H_{2,out}}}{2\;[F_{{\mathrm{CH}}_{4,in}}]}\times 100\%$$
(10)$$\mathrm{CO}\;\mathrm{yield}\;\left(\%\right)=\frac{F_{\mathrm{CO},\;out}}{F_{{\mathrm{CH}}_{4,in}}+F_{{\mathrm{CO}}_{2,in}}}\times 100\%$$
(11)$$\frac{H_{2}}{\mathrm{CO}}=\frac{F_{H_{2,out}}}{F_{\mathrm{CO},\;out}}$$
where $F_{{\mathrm{CH}}_{4}}$, $F_{{\mathrm{CO}}_{2}}$, $F_{H_{2}}$, and $F_{\mathrm{CO}}$ are the flow rates of CH_4_, CO_2_, H_2_, and CO, respectively.

## 3. Results and Discussion

### 3.1. Characterization of Catalysts 

The ceria-zirconia-supported Ni catalysts prepared in this study were characterized using XRD, N_2_ physisorption, H_2_-TPR, CO_2_-TPD, and TGA. 

#### 3.1.1. Crystalline Phase and Catalyst Crystallinity

XRD was used to determine the crystal phases and crystallinities of the catalysts. The XRD diffraction patterns consist of several reflections, with their intensities plotted on the *y*-axis and the measured diffraction angle plotted on the *x*-axis. The results of the XRD analysis for the ZrO_2_-supported Ni catalyst are shown in Figure 1. The XRD pattern of the ZrO_2_ support shows the crystalline phase of a monoclinic structure (m-ZrO_2_) with intense reflections at 2θ = 24.2°, 28.2°, 31.4°, and 34.3° [JCPDS 37-1484]. The obtained diffraction pattern of m-ZrO_2_ is similar to that of Basahel et al. [50] and Sophiana et al. [51].

Meanwhile, the XRD pattern of the mixed metal oxide Ce_0.1_Zr_0.9_O_2_ support is similar to that of the cubic fluorite structure of CeO_2_, without additional reflections corresponding to ZrO_2_ being observed with intense reflections at 2θ = 30.1°, 50.1°, and 60.1°. This indicates the incorporation of ZrO_2_ into the CeO_2_ lattice, leading to the formation of a solid solution, while maintaining the fluorite structure. The diffraction pattern of Ce_0.1_Zr_0.9_O_2_ is similar to that reported by Wolfbeisser et al. [43], Deng et al. [52], and Pham et al. [53]. 

As shown in Figure 1, the Ni formed in the catalyst is in the form of NiO. The three prominent reflections of NiO are observed at 37.28°, 43.30°, and 62.92°, according to JCPDS 73-1523. The intensities of the NiO reflections in the diffraction pattern of the 10%NiO/CeZrO_2_ catalyst are higher than those observed in the diffraction patterns of the other catalysts owing to its higher NiO content. 

#### 3.1.2. Surface Area and Pore Size Distribution of the Catalysts

The BET method was used to determine the mass-specific surface area of each catalyst, while the BJH method was used to determine the pore size distribution of the catalyst. These methods are based on the amount of N_2_ adsorbed and desorbed from the sample. The physical properties of the catalyst in the form of a mass-specific surface area, pore diameter, and pore volume of the synthesized catalyst are listed in Table 1.

It can be observed that the addition of 10 wt.% CeO_2_ to ZrO_2_ increases the surface area and decreases the pore diameter. The ZrO_2_ support has a surface area of 29.19 m^2^ g^−1^ and pore diameter of 18.54 nm. Adding 5 wt.% NiO to the ZrO_2_ support reduces the surface area to 17.23 m^2^ g^−1^ and increases the pore diameter to 22.16 nm. This is due to the addition of NiO, which undergoes sintering and agglomeration during calcination; thus, blocking the pores of the catalyst. Meanwhile, the impregnation of 10 wt.% CeO_2_ in 5%NiO/ZrO_2_ catalyst increases the surface area above that of 5%NiO/ZrO_2_ because 10 wt.% CeO_2_ prevents NiO agglomeration in the catalyst.

Adding 10 wt.% CeO_2_ to ZrO_2_ via co-precipitation to form Ce_0.1_Zr_0.9_O_2_ (CeZrO_2_) can increase the surface area by approximately 30% to 39.83 m^2^ g^−1^ with a pore diameter of 18.54 nm. Addition of 5 wt.% NiO and 10 wt.% NiO to the CeZrO_2_ support decreases the surface area of the catalyst and increases the pore diameter. As more NiO is impregnated into the support, the surface area of the catalyst decreases, and the pore diameter increases. This is due to the addition of NiO, which undergoes sintering and agglomeration during calcination of 700 °C. Meanwhile, nickel sintering occurs at a temperature of 700 °C [31]. This sintering process can make metal agglomerate and block the catalyst’s pores [54].

Catalyst Surface Area (via the BET Method)

The surface area of the catalyst affects the internal diffusion rate, where the reactants are physically adsorbed through the catalyst pores toward the active center of the catalyst. The adsorption–desorption isotherms of N_2_ for the ZrO_2_- and CeZrO_2_-supported NiO catalysts are shown in Figure 2. The isotherms were analyzed using categories created by the International Union of Pure and Applied Chemistry (IUPAC) for the analysis of pores. Based on the plotted results of the volume of N_2_ adsorbed and desorbed versus relative pressure, there are six types of isotherm classifications that can be made.

In Figure 2A, the NiO catalyst supported on ZrO_2_ exhibits a type IV isotherm, corresponding to the classification of mesoporous solids, where this classification describes monolayer-multilayer adsorption on the mesoporous walls. Mesoporous solids have pore diameters in the range of 2–50 nm. In this phenomenon, loop hysteresis occurs because of capillary condensation in the mesopores. In Figure 2B, the isotherm of the CeZrO_2_-supported NiO catalyst is also type IV. The N_2_ physisorption results for the CeZr sample also agreed with the results reported by Deng et al. [52] and Singha et al. [25].

For the NiO catalyst supported by ZrO_2_ and CeZrO_2,_ it can be seen that the adsorbed gas at *P/P*_0_ = 0 is very small, and the monolayer region is not full. At *P/P*_0_ < 0.1, gas adsorption begins to saturate the “monolayer”, even though the amount adsorbed is still very small. At *P/P*_0_ = 0.1–0.4, multilayer adsorption starts to occur, but the amount of N_2_ adsorbed on the CeZrO_2_-supported catalyst was less than that adsorbed on the ZrO_2_-supported catalyst. The type of hysteresis loop observed isa type H_2_(b), which does not show limiting adsorption at high *P/P*_0_. Desorption of this isotherm type contains areas associated with hysteresis closure owing to the stress strength.

Catalyst Pore Size Distribution (via the BJH Method)

The pore size distribution profiles of the ZrO_2_- and CeZrO_2_-supported NiO catalysts under isothermal conditions are shown in Figure 3. Figure 3A shows that the size of the pores in the bare ZrO_2_ support range from 2 to 45 nm, with an average pore size of 18.54 nm and a total pore volume of 0.135 cm^3^ g^−1^. The addition of 5wt.% NiO to the ZrO_2_ support widens the pore size range when compared with that of the bare ZrO_2_, with a larger average pore diameter of 22.16 nm and total pore volume of ca. 0.095 cm^3^ g^−1^ being obtained for the supported catalyst. This larger pore size is probably caused by the calcination carried out twice, which causes NiO and ZrO_2_ to agglomerate and aggregate. Meanwhile, CeO_2_ impregnation in the 5%NiO/ZrO_2_ catalyst results in a narrow pore size distribution, with a smaller average pore size of 7.73 nm and total pore volume of ca. 0.047 cm^3^ g^−1^. This is due to the presence of a CeO_2_ component that fills the pores of the ZrO_2_ support, resulting in a decrease in the pore diameter and volume of the catalyst.

In Figure 3B, it can be seen that the sizes of the pores in the bare CeZrO_2_ support range from 2 to 30 nm, with an average pore size of 13.14 nm and a total pore volume of 0.131 cm^3^ g^−1^. The addition of 5 wt.% NiO to CeZrO_2_ support results in a narrower pore size distribution range than that of the bare CeZrO_2_, with an average pore diameter of 14.78 nm and a total pore volume of 0.112 cm^3^ g^−1^. Meanwhile, the addition of 10 wt.% NiO to the CeZrO_2_ support had a more comprehensive pore distribution range than the bare CeZrO_2_, with an average pore diameter of 16.10 nm and a total pore volume of 0.116 cm^3^ g^−1.^ This larger pore size is probably caused by the calcination being carried out twice, which causes NiO and CeZrO_2_ to agglomerate.

#### 3.1.3. Catalyst Reducibility

H_2_-TPR was performed to determine the level of reduction (i.e., reducibility) of the catalyst and the ability of H_2_ to diffuse into the catalyst. The H_2_-TPR study is important because it informs on how to carry out the activation of the catalyst prior to the catalyst performance evaluation since the synthesized catalyst is in the metal oxide form, whereas the active phase required for the DRM is metallic Ni (Ni^0^). Catalyst reduction using H_2_ produces H_2_O as the main gaseous product. The H_2_-TPR results for the catalysts show that H_2_ consumption occurs at temperatures between 200 and 840 °C. The H_2_-TPR profiles of the NiO catalysts supported on ZrO_2_ and CeZrO_2_ are shown in Figure 4.

The reduction initiation temperatures of the ZrO_2_, 5%Ni/ZrO_2_, and 5%Ni-10%CeO_2_/ZrO_2_ samples are approximately 217, 238, and 200 °C, respectively. Meanwhile, the reduction initiation temperatures of the CeZrO_2_, 5%Ni/CeZrO_2_, and 10%Ni/CeZrO_2_ samples are approximately 250, 209, and 215 °C, respectively. The Ni catalyst exhibits a first reduction peak at approximately 400 °C, which indicates the reduction of NiO to Ni^0^. The second peak in the H_2_-TPR profile of the CeZrO_2_-supported Ni catalyst indicates the reduction of CeO_2_. The addition of CeO_2_ to ZrO_2_ to form CeZrO_2_ can increase the OSC and redox properties of the mixed metal oxide. Among the synthesized catalysts, the 10%Ni/CeZrO_2_ catalyst has the most intense peak for Ni reduction at a low temperature (399 °C). This is due to the higher amount of NiO and shift of the reduction peak to a lower temperature caused by the spillover effect of H_2_. This good redox properties of the CeZrO_2_ support allow for a more effective transfer of oxygen species [42].

#### 3.1.4. Basicity of the Catalysts

The chemical nature of the catalyst can affect its performance in the DRM. Catalysts with strong Lewis basic sites can facilitate the adsorption of acidic CO_2_ molecules and assist in carbon gasification to prevent carbon deposition during the DRM [38]. Meanwhile, a catalyst with strong Lewis acid sites can direct the reaction toward carbon deposition [55].

CO_2_-TPD was performed to determine the basicity of each catalyst by measuring the number of absorbed CO_2_ molecules on the surface. If CO_2_ is mostly desorbed at high temperatures, then the basicity of the catalyst is high because CO_2_ is an acidic probe. However, if CO_2_ is desorbed at low temperatures, then the basicity of the catalyst is low. The desorption of CO_2_ from the catalyst can be divided into three parts: At temperatures between 100 and 200 °C (first section), 250 and 420 °C (second section), as well as 580 and 760 °C (third section), indicating weak, medium, and strong basic sites on the surface of the DRM catalyst. The desorption peak of the catalyst, indicating weak basic sites, is related to the adsorption of CO_2_ by Lewis acid-base pairs [56]. The results of CO_2_-TPD for the ZrO_2_- and CeZrO_2_-supported NiO catalysts are shown in Figure 5.

Figure 5A shows the CO_2_ desorption profile of the ZrO_2_-supported NiO catalyst. Desorption of CO_2_ on the bare ZrO_2_ support shows that desorption occurs in the first and second sections, indicating the presence of weak and moderate alkaline sites, respectively, where the desorption peaks at moderate temperatures are more intense, implying that the catalyst had a higher number of moderate alkaline sites than weak basic sites. The catalyst with the addition of 5 wt.% NiO gave results that are not too different from those of the bare ZrO_2_ support, where desorption occurs in the first and second sections, indicating the presence of weak and moderate alkaline sites, where the desorption peak at the medium temperatures is more intense, implying that these sites are dominant on the catalyst surface. Meanwhile, the presence of 10 wt.% CeO_2_ in the 5%NiO/ZrO_2_ catalyst gave different results, where the desorption peaks at the low and medium temperatures are almost the same, indicating that neither the weak or moderate base sites are dominant.

Figure 5B shows the desorption profile of CO_2_ from the NiO catalyst supported on CeZrO_2_. Desorption of CO_2_ on the bare CeZrO_2_ support shows that desorption occurs in the first and second sections, indicating weak and moderate alkaline sites, where the desorption peaks at low temperatures were more intense, implying that this catalyst has a higher number of weak basic sites than moderate basic sites. Catalysts with the addition of 5 wt.% and 10 wt.% NiO show results that are similar to those of the CeZrO_2_ support, where the desorption peak at a moderate temperature is more intense, indicating that this catalyst predominantly has moderately alkaline sites. Overall, the synthesized catalysts show desorption in the first and second sections, indicating the presence of weak and moderate basic sites. The low CO_2_ desorption temperature (approximately 99 °C) can be attributed to weakly physisorbed CO_2_ [57].

### 3.2. Activity and Stability of the Catalysts

Catalysts are needed for the DRM to maximize syngas production by increasing the rate of the chemical reaction [13]. At an operating temperature of 700 °C and atmospheric pressure, the maximum equilibrium conversion, product yield, and H_2_:CO ratio that can be achieved, according to thermodynamic calculations, were based on the simulation results of Aspen Plus v.10 with the Peng–Robinson equation of state using the RGIBBS reactor model and a CO_2_:CH_4_ ratio of 1, as well as on the calculation performed by Wang et al. [58]. In this study, commercial catalysts were also evaluated for comparison, in addition to evaluating the performance of the synthesized catalysts. The commercial catalysts used were the methanation (containing 28 wt.% Ni, 60 wt.% Al, and 12 wt.% Ca) and steam reforming catalysts (containing 25 wt.% Ni, 58 wt.% Al, 1 wt.% Si, 14 wt.% Ca, and 1 wt.% K). 

Figure 6 shows the conversion profiles of CH_4_ and CO_2_ for the synthesized and commercial catalysts. The 5%Ni/ZrO_2_ catalyst with no added CeO_2_ exhibits an average conversion of CH_4_ and CO_2_ of ca. 58% and 42%, respectively. The impregnation of 10 wt.% CeO_2_ into the 5%Ni/ZrO_2_ catalyst increases the catalyst activity, with average conversions of CH_4_ and CO_2_ of ca. 72% and 56%, respectively. The 10%Ni/CeZrO_2_ catalyst shows higher activity, with average conversions of CH_4_ and CO_2_ of ca. 74% and 55%, respectively, when compared with the 5%Ni/CeZrO_2_ catalyst, with an average conversion of CH_4_ and CO_2_ of ca. 70% and 54%, respectively. This implies that having more Ni in the catalyst improves catalytic performance possibly due to the higher number of active sites per mass of Ni. Meanwhile, the commercial steam reforming catalyst exhibits the highest activity, with average conversions of CH_4_ and CO_2_ of ca. 85% and 65%, respectively. However, when using the commercial methanation catalyst, the activity decreases significantly at the 40^th^ min of the experiment due to catalyst deactivation via carbon deposition. The average CH_4_ and CO_2_ conversions during the performance evaluation over 240 min for the commercial methanation catalyst is ca. 23% and 18%, respectively.

Figure 7 shows the H_2_ and CO yields for the synthesized and commercial catalysts. The 5%Ni/ZrO_2_ catalyst achieves average H_2_ and CO yields of ca. 36% and 39%, respectively. The impregnation of 10 wt.% CeO_2_ into the 5%Ni/ZrO_2_ catalyst increases the yields of H_2_ and CO, with averages of ca. 48% and 44%, respectively. The 10%Ni/CeZrO_2_ catalyst achieves average H_2_ and CO yields of ca. 51% and 43%, respectively, when compared with the 5%Ni/CeZrO_2_ catalyst, with yields of H_2_ and CO of ca. 46% and 43%, respectively.

The combination of CeO_2_ and ZrO_2_ in the CeO_2_-ZrO_2_ support results in better activity than that of ZrO_2_ alone, where the 5%Ni/CeZrO_2_ catalyst exhibits 20–30% higher activity than the 5%Ni/ZrO_2_ catalyst. Meanwhile, the commercial steam reforming catalyst achieves a higher CH_4_ conversion than the synthesized catalysts. This is because the number of active Ni sites in the commercial steam reforming catalyst is greater than that in the synthesized catalysts; therefore, more reactants are converted over the commercial steam reforming catalyst. The commercial steam reforming catalyst has an Ni content of 25 wt.%, whereas the synthesized catalysts have an Ni content of 10 wt.%. This is due to the improved oxygen storage and mobility in the CeZrO_2_ support than in the CeO_2_ and ZrO_2_ alone [59]. Based on the conversion value of the reactants per mass of active Ni in the catalyst, the 10%Ni/CeZrO_2_ catalyst has conversion values of CH_4_ and CO_2_ that are 2.2 and 2.1 times higher than those of commercial steam reforming catalyst.

For the commercial steam reforming catalyst, the average yields of H_2_ and CO products are ca. 66% and 54%, respectively, with the H_2_ yields gradually decreasing throughout the duration of the experiment, although the CH_4_ conversions are stable. This is due to the side reaction of carbon formation by the decomposition reaction of methane (CH_4_ ⇄ C + 2H_2_), which occurs at temperatures above 640 °C in addition to the DRM [31]. At 700 °C, the formation of carbon occurs more spontaneously via the methane decomposition reaction than via the Boudouard reaction based on the change in the Gibbs free energy (ΔG) of the two reactions. The methane decomposition reaction tends to be more dominant in the steam reforming and methanation catalysts. Meanwhile, the yield of H_2_ tends to decrease throughout the duration of the experiment when compared with that of CO. This is due to the reverse WGSR (CO_2_ + H_2_ ⇄ CO + H_2_O), which consumes H_2_ and forms CO.

Among the catalysts evaluated in this study, the synthesized catalysts exhibit better catalytic performance when compared with the commercial catalysts in terms of the conversion of reactants per mass of the active Ni in the catalyst, which is 2.2 times higher for the 10%Ni/CeZrO_2_ catalyst than for the commercial steam reforming catalyst. In addition, based on the H_2_:CO product ratios shown in Figure 8, the 10%Ni/CeZrO_2_ catalyst achieves the highest yields, with an H_2_/CO ratio of ca. 1.4, whereas the thermodynamic ratio is ca. 1.5. The 5%Ni/CeZrO_2_ and 5%Ni-10%CeO_2_/CeZrO_2_ catalysts achieve H_2_:CO ratios of ca. 1.2. Meanwhile, for the catalysts that do not contain CeO_2_, the H_2_/CO ratio is ca. 1, which is lower in comparison.

Overall, the synthesized catalysts have a more stable yield than the commercial catalysts. At the beginning of the experiment, the commercial steam reforming catalyst has a H_2_:CO ratio of ca. 1.6, which decreases throughout the experiment to 1. Meanwhile, for the commercial methanation catalyst, the decrease in the H_2_/CO ratio was more significant, starting from 1.8 and decreasing to 0.6 during the course of the experiment. The average H_2_:CO ratios for the commercial steam reforming and methanation catalysts over the 240-min duration were ca. 1.3 and 0.6, respectively. The decrease in the H_2_:CO ratio is caused by carbon deposition, which deactivates the catalyst. Meanwhile, the 10%Ni/CeZrO_2_, 5%Ni/CeZrO_2_, and 5%Ni-10%CeO_2_/ZrO_2_ catalysts have CeO_2_ components that help minimize the coking process over the catalysts [40].

### 3.3. Carbon Formation in the Spent Catalyst

TGA was used to measure the carbon deposited on each catalyst after the 240-min stability evaluation at a temperature of 700 °C and atmospheric pressure. This method is based on the reduction in the sample mass, which took place under a flow of air as the temperature increases. The mass reduction of the sample occurs because of the evaporation of water and combustion of the carbon in the sample to produce CO_2_.

The TGA profiles obtained for the catalysts evaluated in the DRM (Figure 9) show that the commercial steam reforming and methanation catalysts experience a higher reduction in sample mass due to the combustion of the carbon that was deposited during the catalyst performance evaluation when compared with the synthesized catalysts.

Figure 9 shows the mass reduction profiles of the synthesized and commercial steam reforming catalysts used for 240 min, as well as the profile for the commercial methanation catalyst that was only used for 140 min because the catalyst underwent deactivation. From the profiles, it can be seen that the masses of the synthesized catalysts do not experience a significant reduction when compared with that experienced by the commercial catalysts. The commercial steam reforming and methanation catalysts experience a mass reduction between 600 and 800 °C, which indicates the combustion of the previously deposited carbon on the catalyst. The amounts of carbon deposited on the catalysts are shown in Figure 10.

It can be seen that the amounts of carbon present in the commercial catalysts are much higher than those in the synthesized catalysts. The amount of carbon deposited on the commercial steam reforming catalyst is 0.347 gC/gcat after the stability evaluation during the DRM over 240 min. Meanwhile, on the commercial methanation catalyst, the amount of carbon deposited is 0.386 gC/gcat after the stability evaluation during DRM over 140 min. The amounts of carbon in the commercial catalysts are probably caused by the sintering of Ni and Al_2_O_3_, which causes the aggregation of Ni particles, resulting in enhanced carbon deposition on the catalyst [60]. The higher the amount of Ni in the catalyst, the more carbon is deposited on the catalyst. A large amount of Ni in the commercial steam reforming catalyst (25 wt.%) also reduces Ni dispersion, thereby increasing the chances of sintering [61]. To confirm the occurrence of sintering in the commercial catalysts, further analyses, such as high-resolution transmission electron microscopy, XRD, and/or N_2_ physisorption, are necessary.

The amounts of carbon deposited after the 240-min stability evaluation on the 10%Ni/CeZrO_2_, 5%Ni/CeZrO_2_, 5%Ni/ZrO_2_, and 5%Ni-10%CeO_2_/ZrO_2_ are ca. 0.001 gC/gcat, 0.009 gC/gcat, 0.008 gC/gcat, and 0.006 gC/gcat, respectively. Due to the small amounts of carbon deposited on the synthesized catalysts, carbon deposition on these catalysts can be neglected. The higher stability of the synthesized catalysts when compared with the commercial catalysts is probably due to the catalyst components CeO_2_ and ZrO_2_. The chemical composition of the catalyst with a ceria-zirconia component has a strong influence on the relative contribution of CH_4_ or the CO_2_ activation pathway to carbon formation, as well as on the reactivity of carbon with oxygen species [62].

The vacant oxygen sites on the DRM catalyst provide sites for CO_2_ activation and the breakdown of C-O bonds, thereby reducing the chance of catalyst deactivating due to carbon accumulation. This is because the lattice oxygens on the catalyst can oxidize the carbon to CO [15].

## 4. Conclusions

The performance of the ceria-zirconia-supported Ni catalysts in the DRM was evaluated based on the effect of the addition of ceria, catalyst preparation method, and Ni content on the physicochemical characteristics, performance, and stability of the resultant catalysts. The results of the BET analysis showed that the addition of 10 wt.% of ceria to ZrO_2_ via co-precipitation to form Ce_0.1_Zr_0.9_O_2_ (CeZrO_2_) can increase the surface area by ca. 30%. As more NiO was impregnated into the support, the surface area of the catalyst decreased and the pore diameter increased. The H_2_-TPR results showed that the 10%NiO/CeZrO_2_ catalyst had a more intense peak for NiO reduction at a lower temperature when compared with those of the other synthesized catalysts. This is due to the higher amount of NiO and the shift of the reduction peak to a lower temperature caused by the H_2_ spillover effect. Furthermore, this is buffered with the good redox properties of the support, which allow a more effective transfer of oxygen species. The results of the activity and stability evaluations of the catalysts at a temperature of 700 °C, pressure of 0.92 atm, and WHSV 60,000 mL g^−1^ h^−1^ showed that the 10%Ni/CeZrO_2_ catalyst was the superior one, with a conversion of CH_4_ and CO_2_ of 74% and 55%, respectively. The H_2_:CO ratio in the product was ca. 1.4, indicating a higher formation of H_2_. The amounts of carbon deposited in the synthesized catalysts are minimal, i.e., <0.01 gC/gcat; therefore, the carbon deposition in the synthesized catalysts can be neglected. The use of CeO_2_ and ZrO_2_ in the DRM could help in minimizing coke formation due to the improved oxygen storage and mobility.

## Figures and Tables

**Figure 1 nanomaterials-12-01556-f001:**
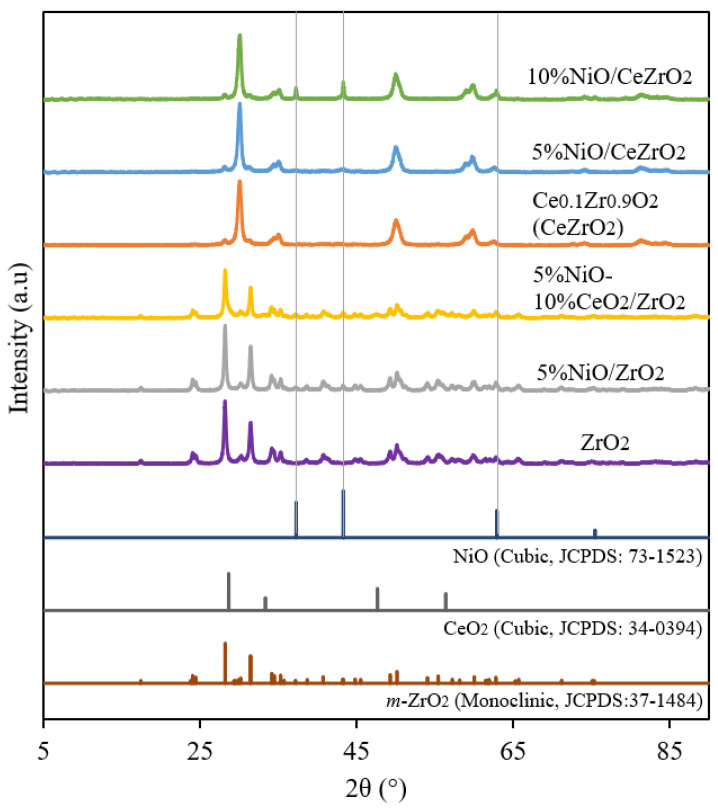
XRD pattern on a ceria-zirconia-supported nickel catalyst.

**Figure 2 nanomaterials-12-01556-f002:**
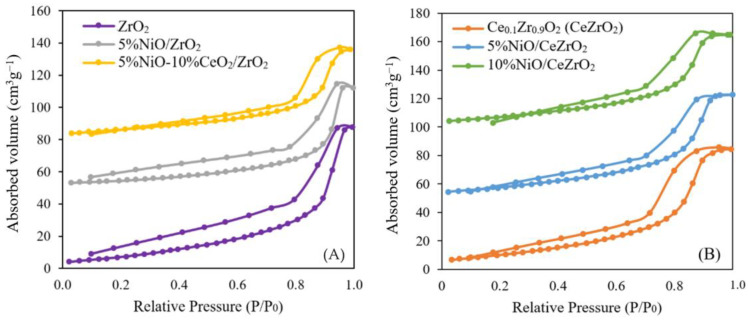
Nitrogen adsorption–desorption of catalyst: (**A**) ZrO_2_-supported NiO catalyst and (**B**) CeZrO_2_-supported NiO catalyst.

**Figure 3 nanomaterials-12-01556-f003:**
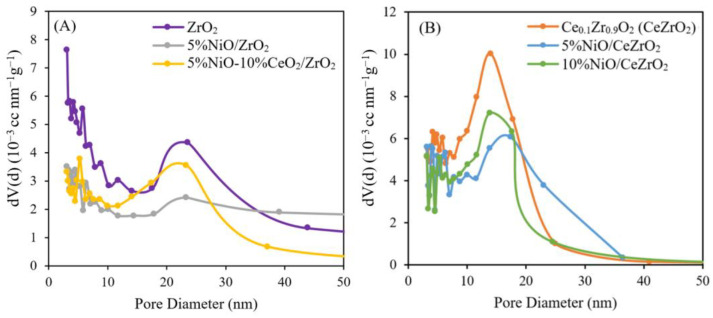
Pore distribution of catalyst: (**A**) ZrO_2_-supported NiO catalyst and (**B**) CeZrO_2_-supported NiO catalyst.

**Figure 4 nanomaterials-12-01556-f004:**
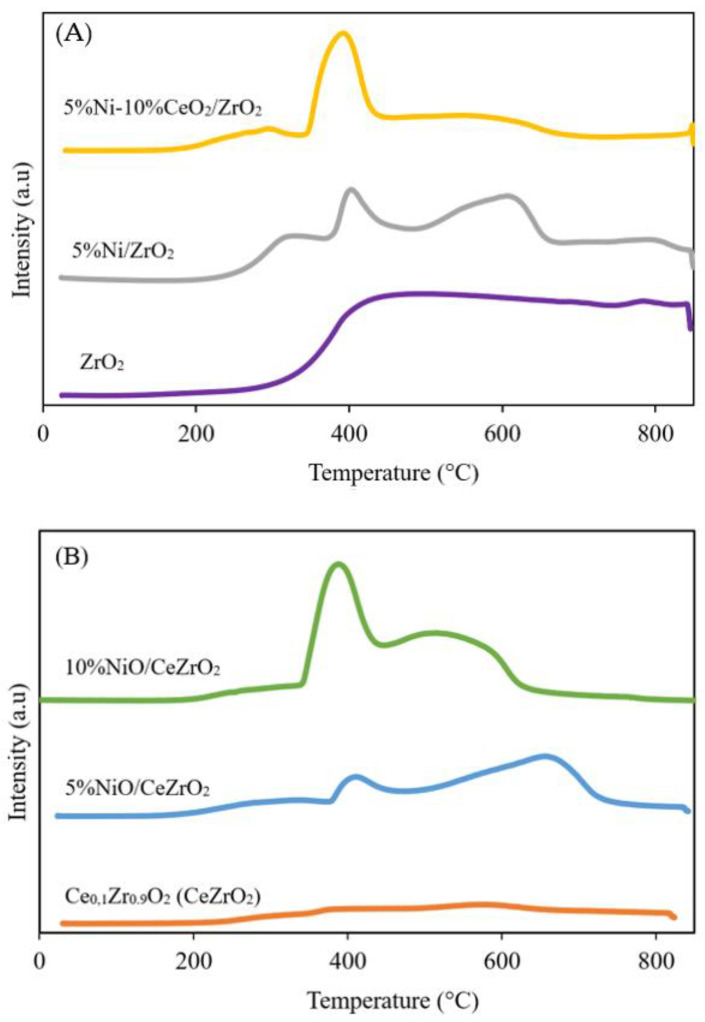
H_2_-TPR profiles of catalyst: (**A**) ZrO_2_-supported Ni catalyst and (**B**) CeZrO_2_-supported Ni catalyst.

**Figure 5 nanomaterials-12-01556-f005:**
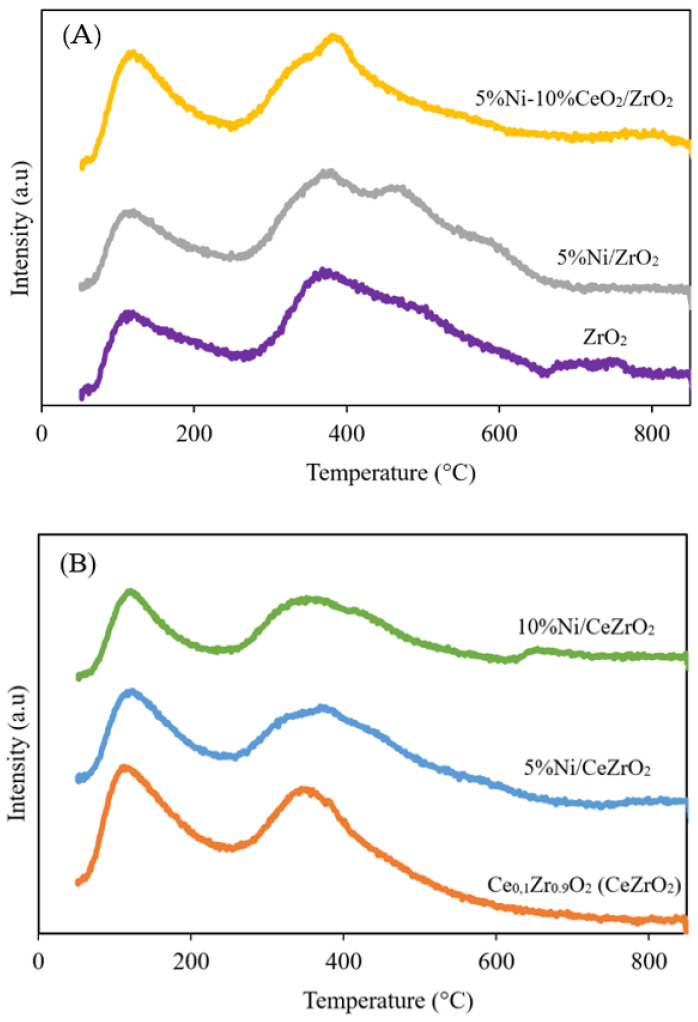
CO_2_-TPD profiles of catalyst: (**A**) ZrO2-supported Ni catalyst and (**B**) CeZrO2-supported Ni catalyst.

**Figure 6 nanomaterials-12-01556-f006:**
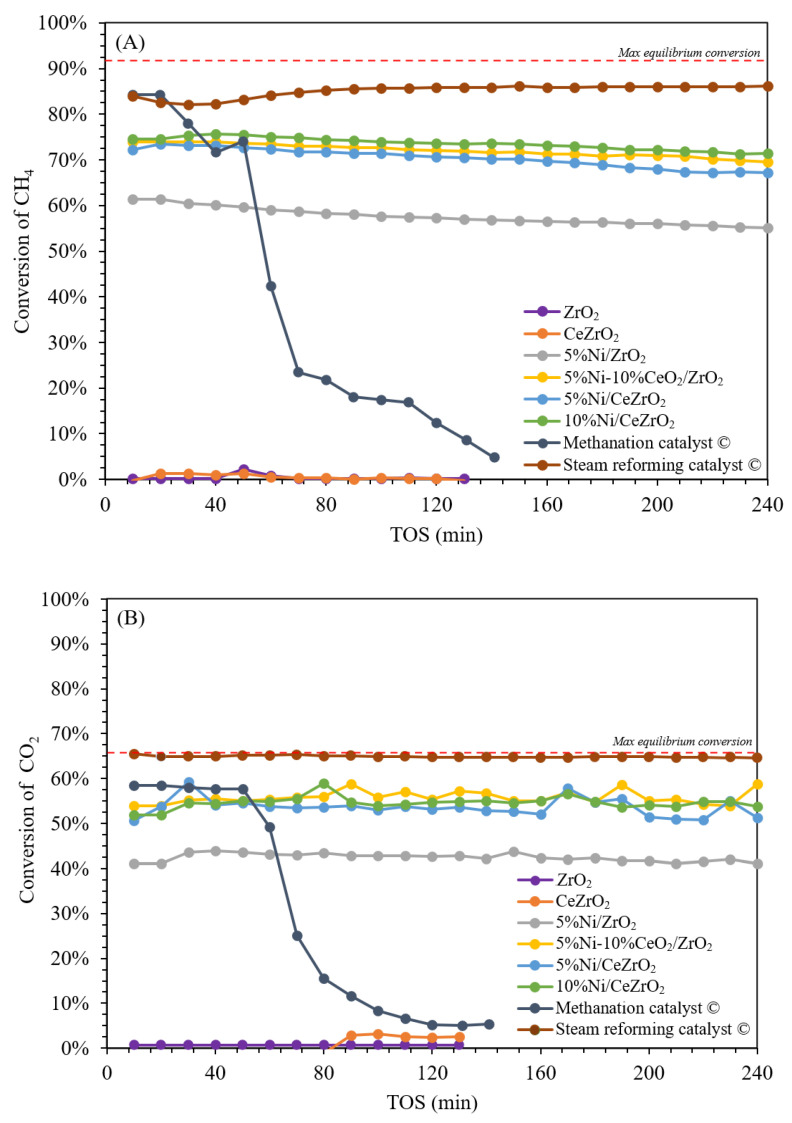
Conversion of reactant on a ceria-zirconia-supported nickel catalyst: (**A**) CH_4_ and (**B**) CO_2_ (temperature of 700 °C, atmospheric pressure, WHSV 60,000 mL g^−1^ h^−1^).

**Figure 7 nanomaterials-12-01556-f007:**
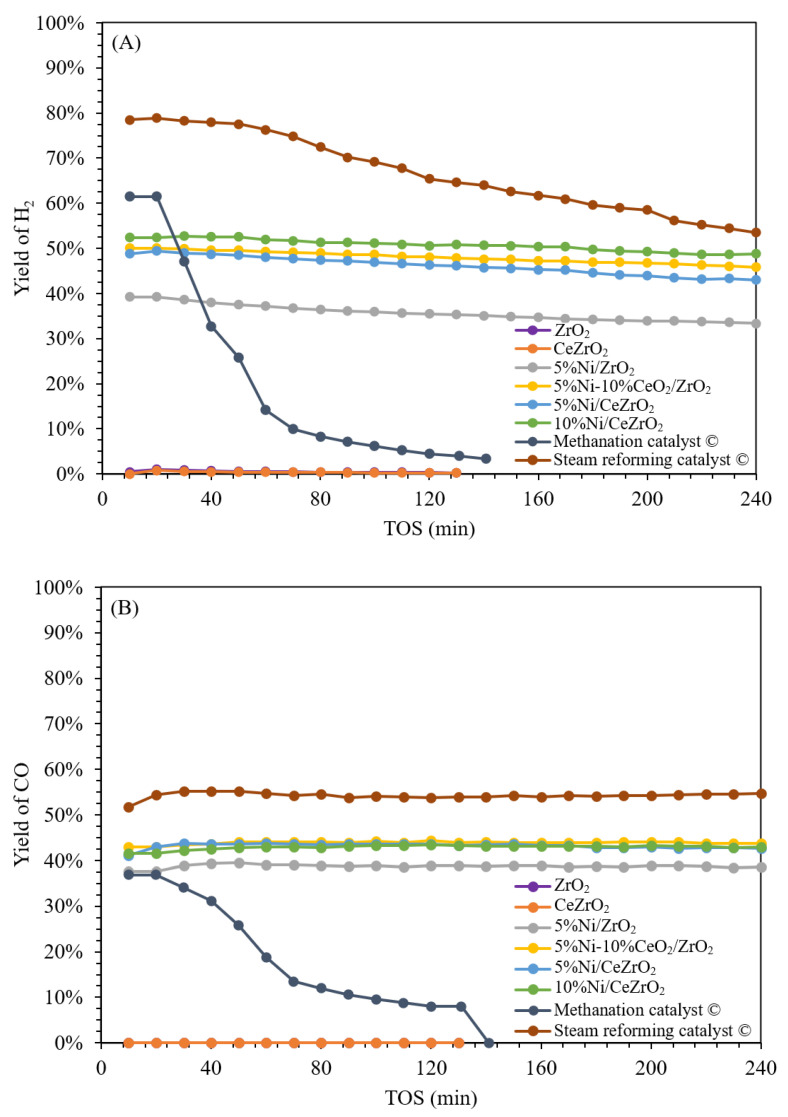
Yield of product on a ceria-zirconia-supported nickel catalyst: (**A**) H_2_ and (**B**) CO (temperature of 700 °C, atmospheric pressure, WHSV 60,000 mL g^−1^ h^−1^).

**Figure 8 nanomaterials-12-01556-f008:**
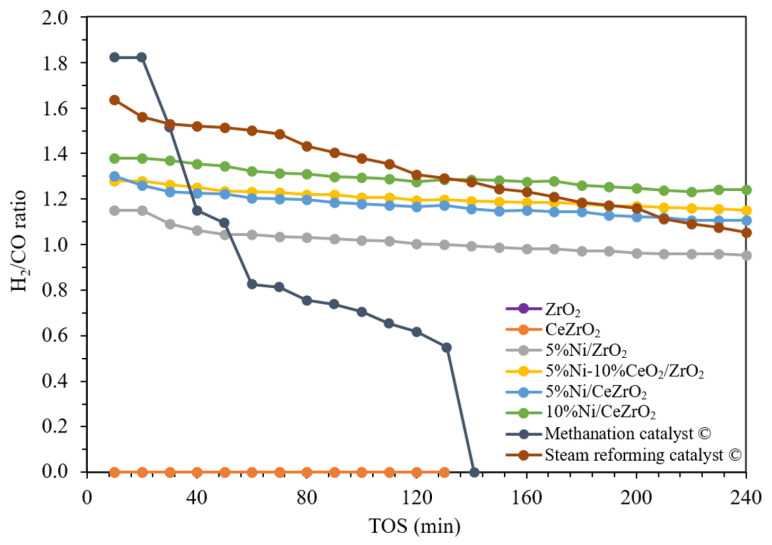
Ratio of H_2_/CO on a ceria-zirconia-supported nickel catalyst (temperature of 700 °C, atmospheric pressure, WHSV 60,000 mL g^−1^ h^−1^).

**Figure 9 nanomaterials-12-01556-f009:**
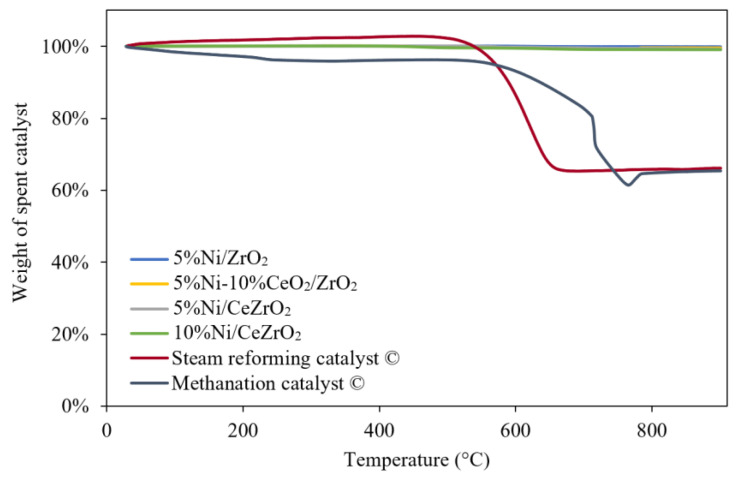
TGA patterns for spent catalyst.

**Figure 10 nanomaterials-12-01556-f010:**
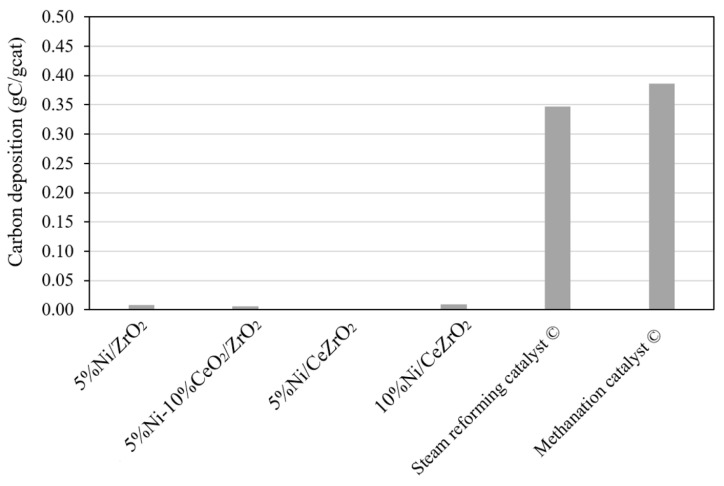
Carbon deposition of spent catalyst during activity and stability test.

**Table 1 nanomaterials-12-01556-t001:** Physical property of ceria-zirconia-supported nickel catalyst.

Catalyst	Surface Area (m^2^ g^−1^)	Pore Diameter (nm)	Total Pore Volume (cm^3^ g^−1^)
ZrO_2_	29.19	18.54	0.135
5%NiO/ZrO_2_	17.23	22.16	0.095
5%NiO-10%CeO_2_/ZrO_2_	24.62	7.73	0.047
Ce_0.1_Zr_0.9_O_2_ (CeZrO_2_)	39.83	13.14	0.131
5%NiO/CeZrO_2_	30.40	14.78	0.112
10%NiO/CeZrO_2_	28.75	16.10	0.116

## Data Availability

The data presented in this study are available on request from the corresponding author.

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
