# Peer review of "Coke-Resistant Ni/CeZrO2 Catalysts for Dry Reforming of Methane to Produce Hydrogen-Rich Syngas"

_nanomaterials, 2022, doi:10.3390/nano12091556_

Round 1
Reviewer 1 Report
Manuscript nanomaterials-1656268, “Coke-resistant Ni/CeZrO2 Catalysts for Dry Reforming of Methane to Produce Hydrogen-rich Syngas” by I. C. Sophiana et al.
The paper deals mainly with the functional activity of Ni/CZO catalyst. First, I do not see much in the manuscript related to Nanomaterial, unless a material with moderate surface area and an obvious mesoporosity. Not a single electronic microscopy image is provided and many statements are highly speculative, being deducted from a poorly characterized materials. For instance, I have serious doubts on the claimed solid solution CZO obtained following the described synthesis method, supported by the clear peak splitting at high angle of figure 1 (despite the poor quality of the patterns) indicating at least a mixture of two solid solution phases. Many sections are too descriptive, for instance lines 212-217, 284-295, …) which may fit in a master thesis but not in a scientific paper. In the N2-physisorption curves (fig 2), the desorption branch is usually considered for the BJH analysis, is collected with few points. The BET model considers generally only 5 points at low pressure, so fig. 2 is more pertinent for the following sections, where pore size distribution is extracted. The section in the lines 250-260 is not supported by evidence, but highly speculative. Therefore, from the material science viewpoint the manuscript is rather poor. Nevertheless, some interesting results can be extracted from the functional characterization as catalyst.
My conclusion is that this manuscript cannot be accepted, mainly because the material is poorly characterized and the several statements are not supported by evidences. I would also suggest submitting, eventually, the improved version to a catalytic journal, because here I do not see any feature of the materials appropriate for a publication under the category “nanomaterials”.
Best regards
Reviewer 2 Report
In this paper, the authors present the synthesis, characterization of the Ni/CeZrO2 catalyst and its activity for dry reforming of methane and compared with a commercial catalyst. The procedure for the synthesis of the different catalyst is clear and the characterization is well described. In addition, the discussion of the characteristics of the prepared materials over the catalytic activity is convincing. However, although the authors compare their catalyst with the commercial one, my main concern is about the nickel content that favours the formation of carbon on the catalyst. Since the concentration of nickel in the commercial is larger than the prepared in the paper, my main doubt is what occurs if the catalyst Ni/CeZrO2 has the same Ni content? I think this experiment should be included in the work.
Consequently, if my main concern is addressed, I believe that the manuscript is suitable for publication in this journal.
Reviewer 3 Report
The work performed by the authors is interesting a well structured; minor revisions are necessary prior to publication.
More comparison should be made with literature data of similar systems (i.e. catalyst(s) with similar composition and/or prepared with different protocols.
Minor points to be addressed: in the figure of the XRD patterns the authors should add the reference file also for the other phases and not just NiO.
For the other graphics (i.e. surface area, pore size, etc.), the authors should use different colours for ZrO2 and 5NiO/CeZrO2, to avoid confusion. Authors should try to be consistent with the colours also for the other hìgraphs.
Round 2
Reviewer 1 Report
The manuscript has been slight modified but the material remain poorly characterized. At least some micrographs on pristine and spent catalyst should be included to support the functional characterization and the conclusions. The support is similar to other Ce-Zr-O materials published in literature, but it is not a single phase; it is most probably a mixture of two Ce-Zr-O phases, at least this should reported. An accurate Rietveld refinement would highlight the mismatch (after better XRD data collection). Overall, the paper would be considered for publication in a catalysis journal, but this decision remain with the Editor.
best regards
Reviewer 3 Report
The authors addressed the points raised by the reviewers. The manuscript can be published in this form without additional changes.
Author Response
We appreciate the positive feedback from the reviewer and we thank the reviewer for all suggestions for improving this manuscript. We hope this manuscript gives a significant contribution to the research on DRM process industrialization.